# Fluorinated Siloxane Modified Layered Double Hydroxide Sealing Film to Enhance the Corrosion Resistance of Anodic Oxide Film of Fricition Stir Welding Joint of Aluminum Alloys

**DOI:** 10.3390/ma15228105

**Published:** 2022-11-16

**Authors:** Wen Li, Tao Wang, Yang Nan, Shao-Jie Li, Wei-Ping Li

**Affiliations:** 1School of Materials Science and Engineering, Beihang University, Beijing 100191, China; 2AVIC Manufacturing Technology Institute, Beijing 100024, China

**Keywords:** aluminum alloys, friction stir welding, anodic oxide film, layered double hydroxide, sealing, corrosion resistance

## Abstract

Aluminum alloys and their welding structures have been widely used in aviation, aerospace, automobiles, ships, and other industrial fields. The non-uniform nature of welding structures of aluminum alloys causes intractable corrosion problems. Anodizing and subsequent sealing processes are common and effective methods to improve the corrosion resistance of welding structures. However, traditional sealing processes like hot water sealing and potassium dichromate sealing are criticized due to energy consumption or toxicity. In this work, a layered double hydroxide (LDH) sealing process with subsequent fluorinated siloxane modification is proposed to improve the corrosion resistance of the anodic oxide film of friction stir welding joints of typical aluminum alloys. The obtained sealing film with typical lamelliform structures of LDH grows well at the defects of oxidation film and also smoothens the sample surface. The hydrophobicity of the film can separate the corrosive medium from the sample surface and further enhance corrosion resistance. As a result, the corrosion current of the welded sample in 3.5 wt.% NaCl solution plummets about 3~4 orders of magnitude compared to the initial state without anodizing, indicating superior corrosion resistance brought by this method.

## 1. Introduction

In recent decades, aluminum alloys have been widely applied in aviation, aerospace, automobiles, ships, machinery manufacturing, electrical, chemical industry and cryogenic devices, and other industrial fields [1,2,3,4,5,6,7,8]. The development tendency of aluminum alloy materials is to meet high strength and low density, which also leads to poorer weldability than steels [9,10]. As a solution, friction stir welding (FSW) is often used to weld aluminum alloys [11,12,13,14,15]. During the welding process, the welding head spins at high speed and rubs with the workpieces, which can heat and soften the joint parts of the workpieces. FSW possesses the advantages of low residual stress, deformation-free welding parts, realizability of automation, simplicity of equipment, low energy consumption, high efficiency, etc. A typical FSW joint consists of four kinds of non-uniform microstructures with gradient variation [16]: base material (BM), heat-affected zone (HAZ), thermomechanical affected zone (TMAZ), and nugget zone (NZ). Such uneven microstructures of welding joints result in marked poor corrosion resistance compared to the substrate of aluminum alloy workpieces, which is the main technical bottleneck to applying FSW technology in industrial fields [17,18,19].

To improve the corrosion resistance of FSW joints, anodic oxide film sealing is considered a significant and necessary way [20,21]. Therefore, hot water sealing (HWS) and potassium dichromate sealing (PDS) are two commonly used and effective sealing methods [22,23,24,25]. However, HSW is energy intensive and cannot meet the corrosion resistance requirements, while PDS gets the blame for toxic hexavalent chromium. Instead, layered double hydroxide (LDH) sealing has been brought into focus due to economy, environment, and corrosion resistance [26,27,28,29]. The main laminate structure of the LDH is composed of two kinds of metal hydroxides with anions embedded between the layers. The anions are bound to the main laminate by an ionic bond or hydrogen bond [26]. With rough microscopic surface texture and plentiful surface hydroxyl groups, LDH can be modified by organic acid, organic acid salt, silane class, and lipid class to achieve a hydrophobic or super hydrophobic effect. Such an effect can keep an interfacial air layer between the corrosion medium and the sample surface to reduce the contact area and further improve the corrosion resistance of anodic oxide film [30,31,32].

In this work, a perfluororooctyltriethoxysilane (denoted as PF, chemical formula CF_3_(CF_2_)_5_CH_2_CH_2_Si(OC_2_H_5_)_3_) modified LDH sealing film is firstly proposed to enhance the corrosion resistance of anodic oxide film of FSW joint of aluminum alloys. LDH nanosheets can directly grow in the micropores and microcracks of the anodic oxide film to seal these pores and defects. Further, via the polymerization reaction of PF and LDH, hydrophobic groups of CF_3_(CF_2_)_5_CH_2_CH_2_Si(O-)_3_ are dexterously introduced onto the sealing film. In addition, the sample surface becomes smoother after LDH sealing and PF modification. Electrochemical measurements in 3.5 wt.% NaCl solutions also prove the excellent corrosion resistance and low corrosion current induced by the above effects. Such a novel method provides an effective and environmental way to deal with the corrosion protection of aluminum alloys, especially welding structures.

## 2. Materials and Methods

### 2.1. The Preparation of FSW Samples of Aluminum Alloys

Two pieces of aluminum alloy sheet (2A12-T4 and 2A97, respectively) with a thickness of 2 mm and length of 600 mm were welded via FSW. The chemical components of the aluminum alloys are shown in Table 1 and Table 2, respectively.

### 2.2. The Preparation of Anodic Oxide Film (CSA Film)

The anodizing procedure of FSW samples was carried out in anodizing electrolyte composed of citric acid monohydrate (8 g/L) and sulfuric acid (45 g/L), while the anodization voltage, time, and temperature were set at 20 V, 25 min, and 30 °C, respectively. The obtained anodic oxide film is denoted as a CSA film.

### 2.3. The LDH Sealing Process

Via slowly adding 6 g/L of LiOH solution, the pH of Li_2_CO_3_ solution (0.2 mol/L) is adjusted to 12. The anodized samples were immersed in the Li_2_CO_3_ solution at 70 °C for 60 min and then washed with deionized water and dried by cold-blast air. Previous literature has indicated that LDH can be directly formed on the surface of alloys in aqueous solutions under specific compositions and pH values.

### 2.4. The Modification of LDH Sealing Film via PF (C_14_H_19_F_13_O_3_Si)

A mixed solution was prepared by adding 5 g of PF and 30 mL of methanol into 60 mL of deionized water via stirring at 30 °C. The LDH film-sealed aluminum alloy welding samples were immersed in the mixed solutions at 30 °C for 120 min, followed by washing in deionized water and drying under 60 °C for 120 min.

### 2.5. Characterization Structural Analysis

Morphology features of the aluminum alloys were observed by scanning electron microscope (SEM, JEOL 7500, Tokyo, Japan). The roughness of the sample surface was acquired through an atomic force microscope (AFM, ICON, Dublin, Ireland, Veeco, Plainview, NY, USA, Bruker, Billerica, MA, USA). The contact angle tests were carried out via the Attension Theta (Biolin Scientific, Västra Frölunda, Sweden), and the environment temperature and volume of H_2_O were set at 25 °C and 40 µL, respectively.

### 2.6. Electrochemical Measurements

All of the electrochemical measurements were carried out via an electrochemical workstation (CS350H). Saturated calomel electrode (SCE), platinum electrode, and welded aluminum alloy samples were applied as the reference electrode, counter electrode, and working electrodes. The scan rate was set at 10 mV s^−1^ for the polarization curves tests, and electrochemical impedance spectroscopy (EIS) tests were carried out from 100 kHz to 0.01 Hz at 10 mV.

## 3. Results and Discussions

### 3.1. The Characterization of CSA Film

SEM was applied to investigate different areas of the FSW joint section after anodizing to evaluate the homogeneity of the obtained CSA film. As shown in Figure 1, the thickness of CSA film on each joint area (HAZ, NZ, BM-2A12, or BM-2A97) is relatively homogeneous, with a measured deviation of no more than 1.2 μm. The average thickness of CSA film on HAZ is 10.5 μm (Figure 1a), which is 9.7 μm for NZ (Figure 1b). The film thicknesses for BM are relatively larger than HAZ and NZ, which are 12.4 μm for BM-2A12 (Figure 1c) and 11.3 μm for BM-2A97 (Figure 1d). On the whole, the thickness of CSA film is fairly uniform, with values between 9.7~13.5 μm.

AFM was used to further investigate the surface states of the CSA film on different FSW joint areas. The pits on the CSA film on HAZ were caused by the second phase dissolution during anodizing (Figure 2a), while the pits on the CSA film of NZ and BM-2A12 were attributed to the formation of fenestra during anodizing (Figure 2b,c). The raised part in the CSA film of BM-2A97 was induced by the warping of the rupture CSA film after anodic oxidation. The roughness measuring results of CSA film on different FSW joint areas are shown in Table 3. The roughness of CSA film on BM-2A97 is highest with a R_a_ of 71.2 nm and lowest for BM-2A12 (R_a_ = 40.2 nm). The roughness measuring results are consistent with the AFM test results mentioned above.

### 3.2. The Characterization of CSA-LDH Film

To improve the corrosion resistance of CSA film, LDH sealing film was in-situ synthesized after anodic oxidation. Under the alkaline condition, the Al^3+^ ions in the anodizing film would be transformed into Al(OH)_3_ collosol and then react with Li_2_CO_3_ to form LiAl-LDH nanosheets (Li_2_[Al_2_(OH)_6_]_2_CO_3_ ∙ nH_2_O) on the sample surface [33]. The microstructures of the obtained composite film (denoted as CSA-LDH film) in different areas were studied via SEM, and the results are shown in Figure 3. As can be seen from the low magnification images (left), the CSA-LDH film layer is overall smooth, and the cavity defects could be attributed to the dissolution of the second phase particles during the anodic oxidation process. The second phase particles in NZ were refined at high temperatures during the FSW process so that NZ exhibits the flattest surface (Figure 3b). The cavity defects in HAZ (Figure 3a) and TMAZ (Figure 3c) are more apparent, which could be ascribed to the recrystallization and growth of the second phase particles under heat effect. The second phase particles were bulky and sparse in BM-2A12, while tiny and numerous in BM-2A97, which can explain the different surface appearances in Figure 3d,e. The middle images with higher magnification indicate that the CSA-LDH film exists in the whole area, including the cracks and cavities, while some small cavities were even completely sealed. It can be concluded that a continuous CSA-LDH film was formed on the sample surface. At the same time, micro-cracks were found in the CSA-LDH film, which was caused by the film volume change during the sealing process. Zooming in further indicates the CSA-LDH film is composed of a petal-like lamellar structure (Figure 3a–e, right), which is a typical LDH structure. 

### 3.3. The Characterization of CSA-LDH-PF Film

To further enhance the corrosion resistance of the aluminum alloy FSW joints, PF was applied to modify the obtained CSA-LDH film, and the obtained composite film is denoted as the CSA-LDF-PF film. The two-step chemical reactions during the modification process are described in Figure 4. The Si-OC_2_H_5_ group of PF was first hydrolyzed to form the Si-OH group. Then a polycondensation reaction occurred between the Si-OH group and the LDH surface-OH group.

Figure 5 reveals the microstructure of CSA-LDH-PF film in different areas. As indicated by the arrows in Figure 5a–e, the volumetric expansion effect of CSA-LDH-PF film resulted in stress concentration at the tip of the film and the relevant warpage. Under higher magnification, the typical LDH structure is found to be well-reserved after modification by PF. At the same time, CSA-LDH-PF film grows well at the defects and retains partial morphologies of CSA-LDH film. Importantly, lamelliform structures were able to grow in the depth of the cracks and continuously connect the lamelliform structures on both sides of the cracks (viz, cracks sealing), as indicated by the circles in Figure 5e. The degree of surface roughness of the CSA-LDH-PF film on different areas kept consistent with the pattern in Figure 3. Meanwhile, due to the grafting of the CF_3_(CF_2_)_5_CH_2_CH_2_Si(O-)_3_ groups, the CSA-LDH-PF film on different areas is flatter than the CSA-LDH film on corresponding areas.

AFM was applied to further evaluate the surface states of the CSA-LDH-PF film on the different FSW joint areas. As shown in Figure 6, the surface of the CSA-LDH-PF is covered by isolated peaks of various sizes, which can be attributed to the reserved vertical LDH nanosheet structure. The surface roughness of the CSA-LDH-PF film on different FSW joint areas is illustrated in Table 4, which is approximately half of the CSA film, indicating the surface of the sample became smoother after LDH sealing and PF modification. The test results of the AFM and roughness of CSA-LDH-PF film are in accord with the SEM images in Figure 5.

### 3.4. Electrochemical Measurements

To investigate the corrosion resistance of the aluminum alloy sample protected by the CSA-LDH-PF film, polarization curves (seen in Figure 7) were tested in 3.5 wt.% NaCl solutions and the fitting results of the polarization curves are shown in Table 5. There is little distinction between the corrosion potentials (*E*_coor_) of the different samples. Nevertheless, compared to the corrosion current (*i*_coor_) of the Blank aluminum alloy sample without surface treatments (1.267 × 10^−5^ A·cm^−2^), the *i*_coor_ of the sample after anodizing is nearly ten times smaller (1.657 × 10^−6^ A·cm^−2^), while the CSA-LDH film generated by LDH sealing could further decrease the *i*_coor_ to 2.483 × 10^−7^ A·cm^−2^. However, the promotion of corrosion resistance induced by the LDH sealing is still dissatisfactory due to the imperfect sealing effect of the fenestra. During the PF modification process, more cracks were covered or connected by the reaction products of PF and LDH. As a result, the *i*_coor_ of CSA-LDH-PF film is significantly reduced to 7.150 × 10^−9^ A·cm^−2^, indicating superior corrosion resistance.

Impedance spectroscopy was carried out to get more insight into the corrosion kinetics of aluminum alloy samples in 3.5 wt.% NaCl. Not unexpectedly, the capacitive arc radius of the Blank sample is the minimum, while the CSA samples come second (Figure 8a-b). After the LDH sealing and PF modification, the capacitive arc radius further increases significantly. The results of the bode plot are in line with the above analysis. The blank sample exhibits the minimum impedance, while subsequent the anodization, LDH sealing, and PF modification process can significantly increase the impedance, especially in the low-frequency region (Figure 8c). The information on the barrier layers of the samples can be seen from the phase angle plot (Figure 8d). For the blank sample, there is only an original barrier layer indicating the impedance induced by the metal surface. A new barrier layer at around 10^0^ Hz can be attributed to the anodic oxide film after anodization. Further, the LDH sealing film endowed the surface with an extra barrier layer in the lower frequency, and the extra barrier layer can be signally strengthened by PF modification.

Equivalent circuit fitting diagrams (Figure 9) were adopted to accurately estimate the samples’ impedance; the fitting results are shown in Table 6. The film resistance (R_f_) of the CSA-LDH-PF sample is 44583 Ω·cm^2^, which is significantly greater than that of the CSA-LDH (12804 Ω·cm^2^) and CSA (2909 Ω·cm^2^) samples. At the same time, the charge transfer resistance (R_ct_) of the CSA-LDH-PF reaches up to 891,310 Ω·cm^2^, far superior to those of CSA-LDH (28,156 Ω·cm^2^), CSA (3919 Ω·cm^2^) and Blank (852 Ω·cm^2^), revealing that the CSA-LDH-PF film endows the aluminum alloy sample with excellent corrosion resistance.

### 3.5. Contact Angle Tests

The contact angle was measured to verify the hydrophobicity of the CSA-LDH-PF film, and the results are shown in Figure 10. The water drop on the blank sample (after grinding and polishing) exhibits a contact angle of 79.2° (Figure 10a). After anodizing, the surface of the aluminum alloy was replaced by a porous oxidation film, which made the surface hydrophilic and decreased the contact angle to 6.1° (Figure 10b). LiAl-LDH nanosheets (Li_2_[Al_2_(OH)_6_]_2_CO_3_ ∙ nH_2_O) with surface hydroxyl groups and bound water were generated on the oxidation film during the LDH sealing process, which made the surface more hydrophilic (contact angle = 4.6°). After subsequent modification by PF, the surface hydroxyl groups reacted with PF and transformed to hydrophobic CF_3_(CF_2_)_5_CH_2_CH_2_Si(O-)_3_ [34] (seen in Figure 4). The water drop exhibited a large contact angle of 144.6° on the low-surface-energy CSA-LDH-PF film (Figure 10d). The hydrophobicity of the CSA-LDH-PF film is beneficial to separate corrosive medium from the samples and thus further enhances corrosion resistance.

## 4. Conclusions

In summary, a PF-modified LDH sealing film is demonstrated to enhance the corrosion resistance of the CSA film of FSW joints of aluminum alloys. A protective CSA-LDH-PF film is successfully prepared on the FSW joint of aluminum alloys by anodizing, LDH sealing, and PF modification. The obtained CSA-LDH-PF film grows well at the defects and retains partial morphologies of the CSA-LDH film. What is more, the lamelliform structures can grow at the crack depth and continuously seal the crack. The roughness tests indicate that the sample became smoother after the LDH sealing and PF modification. The fitting results of the polarization curves and impedance spectroscopy in 3.5 wt.% NaCl solutions indicate that the CSA-LDH-PF sample exhibits the lowest corrosion current of 7.150 × 10^−9^ A·cm^−2^ and the highest film resistance of 44583 Ω·cm^2^. In addition, the hydrophobicity of the CSA-LDH-PF film is beneficial to separate the corrosive medium from the samples and thus further enhances corrosion resistance. Such a novel sealing technology has the advantages of low cost and environmental friendship and achieves distinguished corrosion resistance. We deem that this method could also be appropriate for the surface protection of other valve metals (like magnesium alloys and titanium alloys) and their welding structures.

## Figures and Tables

**Figure 1 materials-15-08105-f001:**
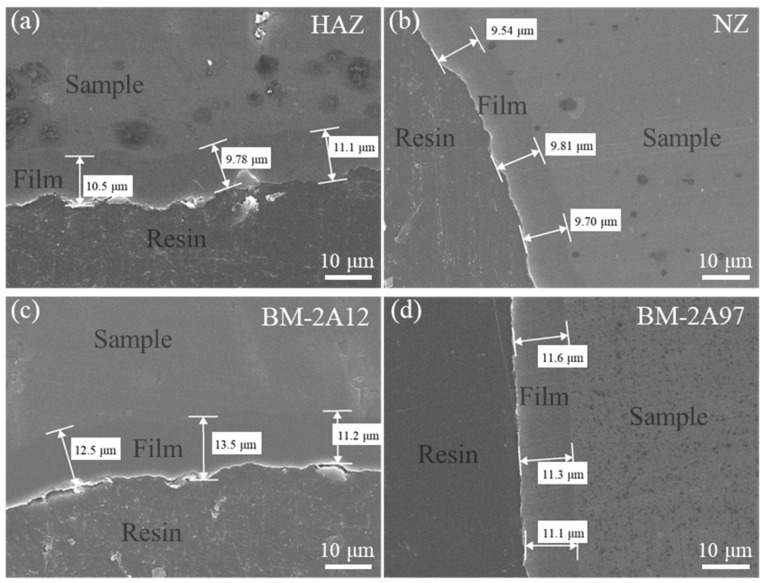
Section SEM images of the FSW joint after anodizing. (**a**) HAZ, (**b**) NZ, (**c**) BM-2A12 and (**d**) BM-2A97.

**Figure 2 materials-15-08105-f002:**
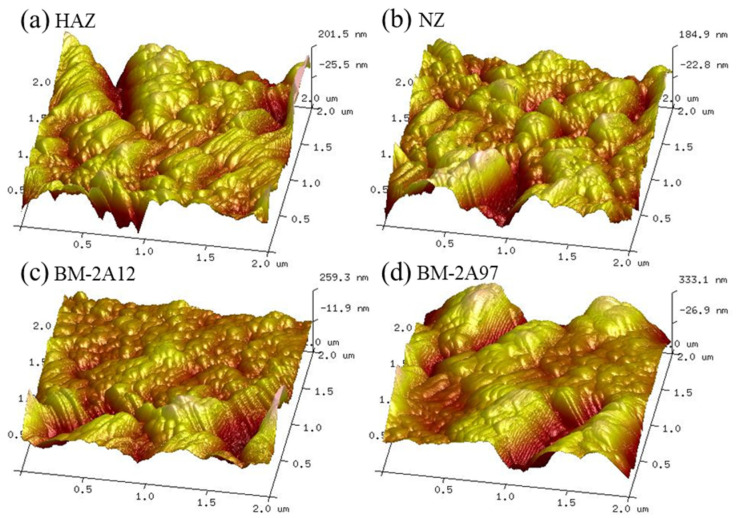
AFM images of the CSA film on different FSW joint areas. (**a**) HAZ, (**b**) NZ, (**c**) BM-2A12 and (**d**) BM-2A97.

**Figure 3 materials-15-08105-f003:**
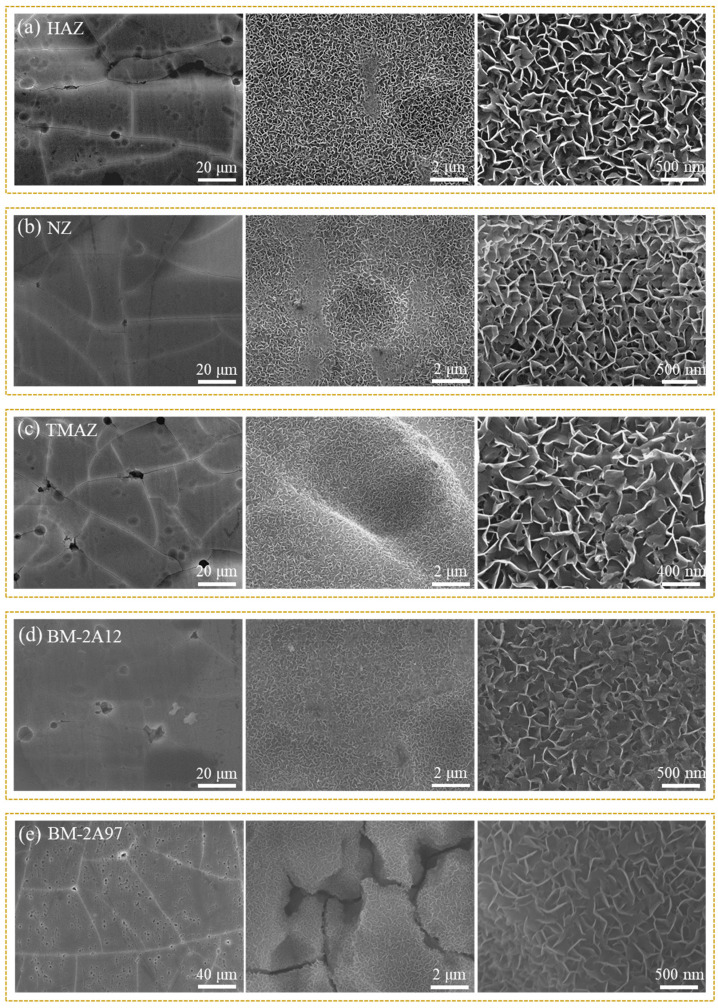
SEM images of the CSA-LDH film on different FSW joint areas. (**a**) HAZ, (**b**) NZ, (**c**) TMAZ, (**d**) BM-2A12 and (**e**) BM-2A97.

**Figure 4 materials-15-08105-f004:**
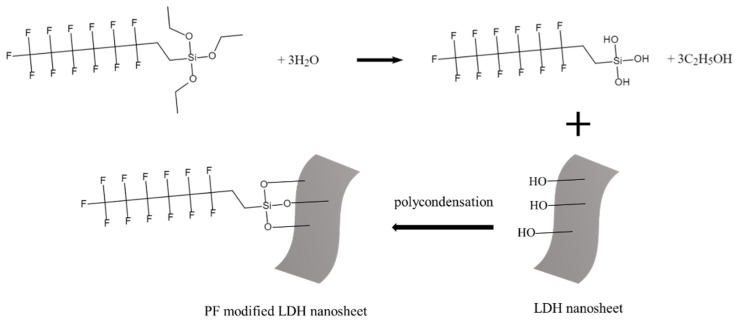
The sketch map of the chemical reactions during the modification process.

**Figure 5 materials-15-08105-f005:**
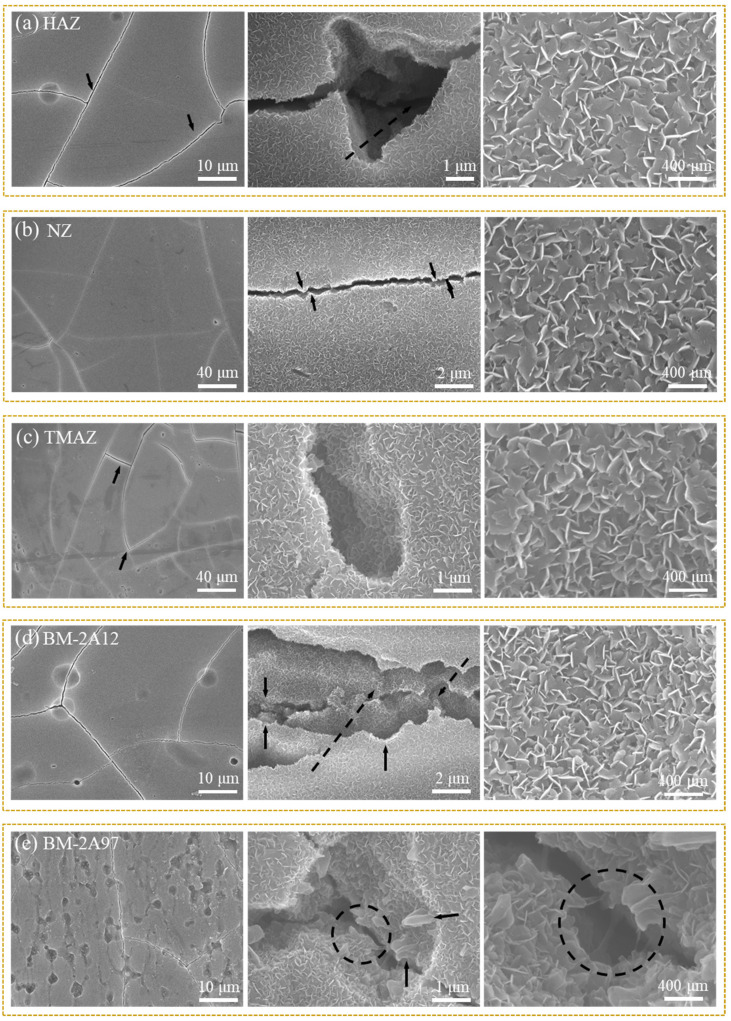
SEM images of the CSA-LDH-PF film on different FSW joint areas. (**a**) HAZ, (**b**) NZ, (**c**) TMAZ, (**d**) BM-2A12 and (**e**) BM-2A97.

**Figure 6 materials-15-08105-f006:**
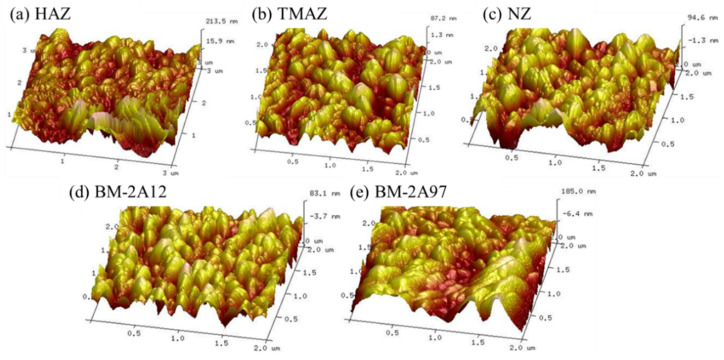
AFM images of CSA-LDH-PF film on the different FSW joint areas. (**a**) HAZ, (**b**) NZ, (**c**) TMAZ, (**d**) BM-2A12 and (**e**) BM-2A97.

**Figure 7 materials-15-08105-f007:**
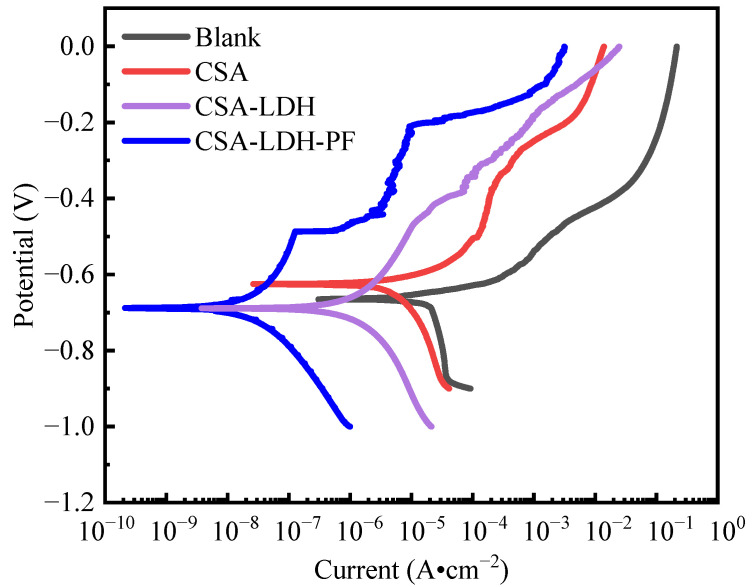
Polarization curves of samples with different anodizing and sealing treatments in 3.5 wt.% NaCl solutions.

**Figure 8 materials-15-08105-f008:**
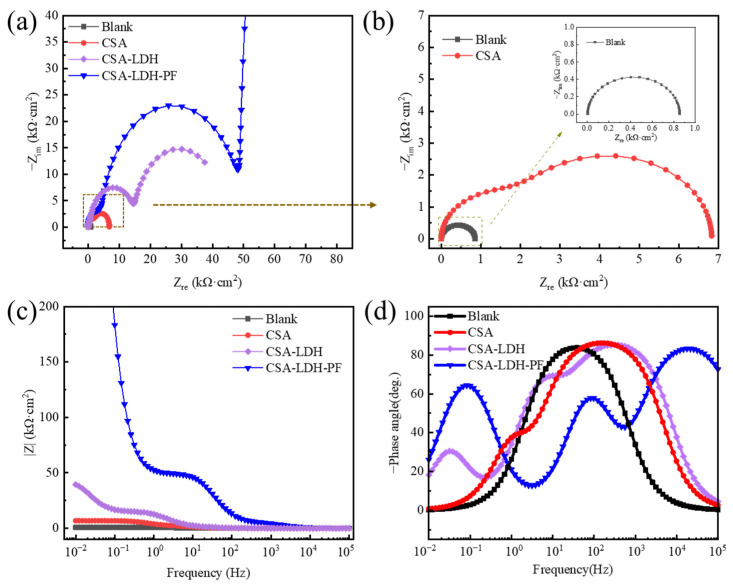
Impedance spectroscopy of the samples with different anodizing and sealing treatments in 3.5 wt.% NaCl solutions. (**a**) Nyquist plot, (**b**) magnified Nyquist plot, (**c**) Bode plot, and (**d**) phase angle plot.

**Figure 9 materials-15-08105-f009:**
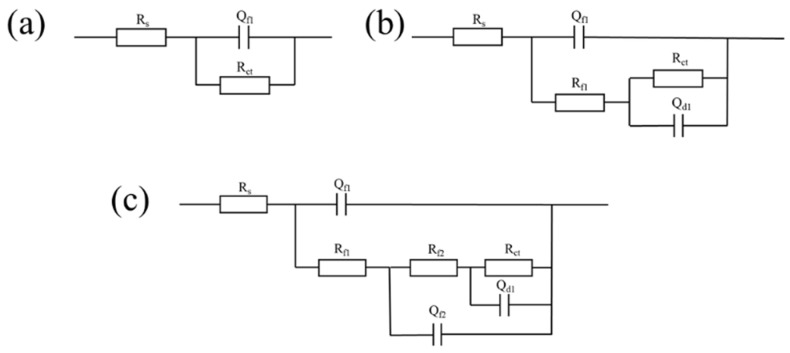
Equivalent circuit fitting diagrams of the (**a**) Blank, (**b**) CSA, and (**c**) CSA-LDH and CSA-LDH-PF samples. R_S_: solution resistance, R_f_: film resistance, Q_f_: capacitance of native oxidized film/passivated film, R_ct_: charge transfer resistance, Q_dl_: double layer capacitance.

**Figure 10 materials-15-08105-f010:**
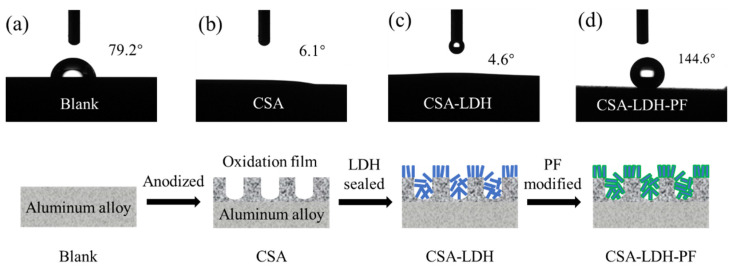
Contact angles of the (**a**) Blank, (**b**) CSA, and (**c**) CSA-LDH and (**d**) CSA-LDH-PF samples.

**Table 1 materials-15-08105-t001:** Chemical composition of 2A12-T4 alloys (wt.%).

Element	Mg	Cu	Mn	Zn	Ti	Cr	Fe	Si	Al
Content	1.40	4.40	0.60	0.048	0.022	0.004	0.180	0.076	Bal.

**Table 2 materials-15-08105-t002:** Chemical composition of 2A97 alloys (wt.%).

Element	Cu	Li	Mn	Mg	Zn	Zr	Al
Content	3.82	1.53	0.30	0.48	0.51	0.16	Bal.

**Table 3 materials-15-08105-t003:** Roughness of the CSA film on different areas of the FSW joint (R_q_: r.m.s roughness, R_a_: mean roughness).

Areas	BM-2A12	BM-2A97	HAZ	NZ
R_q_/nm	59.6	94.0	57.9	54.2
R_a_/nm	40.2	71.2	42.2	41.4
R_a_/R_q_	0.674	0.757	0.729	0.764

**Table 4 materials-15-08105-t004:** Roughness of the CSA-LDH-PF film on different areas of the FSW joint (R_q_: r.m.s roughness, R_a_: mean roughness).

Areas	TMAZ	BM-2A12	BM-2A97	HAZ
R_q_/nm	28.5	25.2	55.3	49.5
R_a_/nm	22.1	20.3	43.9	37.1
R_a_/R_q_	0.775	0.806	0.794	0.749

**Table 5 materials-15-08105-t005:** Fitting results of the polarization curves in Figure 7.

Samples	Blank	CSA	CSA-LDH	CSA-LDH-PF
*E*_corr_ (V)	−0.665	−0.626	−0.687	−0.680
*i*_corr_ (A·cm^−2^)	1.267 × 10^−5^	1.657 × 10^−6^	2.483 × 10^−7^	7.150 × 10^−9^

**Table 6 materials-15-08105-t006:** Fitting results of the impedance spectroscopy in Figure 7.

Samples	R_s_/Ω·cm^2^	R_f1_/Ω·cm^2^	Q_f1_/F·cm^−2^	R_f2_/Ω·cm^2^	Q_f2_/F·cm^−2^	R_ct_/Ω·cm^2^	Q_d1_/F·cm^−2^
Blank	2.577	/	9.25 × 10^−5^	/	/	852	/
CSA	3.188	2909	1.06 × 10^−5^	/	/	3919	6.01 × 10^−5^
CSA-LDH	5.075	12,804	4.04 × 10^−7^	12,518	4.04 × 10^−6^	28,156	2.82 × 10^−4^
CSA-LDH-PF	65.56	44,583	2.26 × 10^−8^	44,061	1.22 × 10^−7^	891,310	9.20 × 10^−6^

## Data Availability

Data will be available upon request through the corresponding author.

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
