# Peer review of "Fluorinated Siloxane Modified Layered Double Hydroxide Sealing Film to Enhance the Corrosion Resistance of Anodic Oxide Film of Fricition Stir Welding Joint of Aluminum Alloys"

_materials, 2022, doi:10.3390/ma15228105_

Round 1

Reviewer 1 Report

The paper presents interesting experimental study with new technology for sealing to enhance the corrosion resistance of welded joint. Please address the following comments.

1. English language needs improvement

2. The significance of this study with possible applications needs to be included in the Conclusion

3. Extend the abstract with additional details.

4. research gap and novelty needs to be written very clearly at the end of Introduction

5. all characterisation techniques need to be given in details in section 2

6. Breakdown results and discussions section into several subsections based on the results presented

7. Improve the discussion on images in Figure 3 and Figure 4 to differentiate the zones with the films. Same comment for Figure 7

Author Response

Thank you very much for your kind comments and good suggestions. They are helpful to improve the quality of our manuscript. The revisions of the manuscript are marked up using the “Track Changes” function, which can be seen in the attachment.

  1. English language needs improvement.

Response: We are sorry for the deficient English expressions in our manuscript. We have carefully checked and revised our manuscript.

  1. The significance of this study with possible applications needs to be included in the Conclusion.

Response: Thank you for your advice. We deem that this method could also be appropriate for the surface protection of other valve metals (like magnesium alloys and titanium alloys) and their welding structures. The relevant description has been added in the Conclusion.

  1. Extend the abstract with additional details.

Response: As you suggested, additional details of anodizing, sealing process and hydrophobicity of the film have been added in the abstract.

  1. Research gap and novelty needs to be written very clearly at the end of Introduction.

Response: As you suggested, we have added the relevant expressions at the end of Introduction.

  1. All characterisation techniques need to be given in details in section 2.

Response: The relevant techniques have been added and given detailly in “2.5 Characterization structural analysis” and “2.6 Electrochemical measurements”.

  1. Breakdown results and discussions section into several subsections based on the results presented.

Response: As you suggested, the results and discussions section has been decomposed into the follow subsections: 3.1 The characterization of CSA film, 3.2 The characterization of CSA-LDH film, 3.3 The characterization of CSA-LDH-PF film, 3.4 Electrochemical measurements and 3.5 Contact angle tests.

  1. Improve the discussion on images in Figure 3 and Figure 4 to differentiate the zones with the films. Same comment for Figure 7.

Response: Thank you for your advice. Additional discussions are added and can be seen in 3.2 The characterization of CSA-LDH film, 3.3 The characterization of CSA-LDH-PF film and 3.4 Electrochemical measurements.

Reviewer 2 Report

Manuscript ID materials-1982123 entitled " Fluorinated siloxane modified layered double hydroxide sealing film to enhance the corrosion resistance of anodic oxide film of fricition stir welding joint of aluminum alloys" for journal of Materials has been reviewed.

The manuscript was interesting and well-motivated. The following list of comments will help to further improve the manuscript:

1- The novelty of the study should be further explained (in introduction…)

2- In introduction section, more references should be added.  (especially about different studies)

3- Tables 3, 4, 5 and 6 should be checked.

4- Figure 1 and 7, the resolution should be increased. (and magnify)

5- “The LDH sealing process”and “The modification of LDH sealing film via PF” …. detail the processes further. (add pictures if possible)

6- Evaluation of SEM images should be increased. (Figure 3 and 4)

7- ….. the CSA-LDH-PF sample exhibits the maximal resistance, and the barrier layer ap-177 pears in the low frequency segment………. More explain, Why?

8-  ….. Hydrophilic CSA-LDH 203 film with surface hydroxyl groups generated during the sealing process further de-204 creased the contact angle to 4.6° (Figure 9c). However, after modification by PF, the 205 surface groups are changed to hydrophobic CF3(CF2)5(CH2)2- with low surface energy 206 [30] and…….. This section should be more detailed. (Specific explanations required)

9- Conclusions section should be enriched.

10- More literature studies should be added to the introduction and other sections (DOIs given below).

DOI-1   https://doi.org/10.1007/s13369-021-06243-w  (about different studies)

DOI-2  https://doi.org/10.35193/bseufbd.1075980  (about different studies)

DOI-3  https://doi.org/10.53525/jster.1174394  (about welding aluminum alloys)

----------------------------------------------------

* It will be ready for publication after the specified corrections.

** I want to see article after the revision.

-----------------------------------------------------

Congratulations to the authors.

I wish the authors success in their future academic studies.

Kind regards.

Author Response

Thank you very much for your kind comments and good suggestions. They are helpful to improve the quality of our manuscript. The revisions of the manuscript are marked up using the “Track Changes” function, which can be seen in the attachment.

  1. The novelty of the study should be further explained (in introduction…)

Response: Thank you for your advice. The relevant statements have been added in the end of introduction.

  1. In introduction section, more references should be added.  (especially about different studies)

Response: As you suggested, References 3, 9 and 10 are added.

  1. Tables 3, 4, 5 and 6 should be checked.

Response: We are sorry for the careless spelling mistakes in Tables 3, 4 and 6. The above mistakes have been revised in our manuscript. At the same time, the definition of Ecorr(V) is added in the discussion of Table 5 to avoid unnecessary puzzle.

  1. Figure 1 and 7, the resolution should be increased. (and magnify)

Response: We are sorry for the unclear figures and have updated the relevant figures (Figure 1 and Figure 8).

  1. “The LDH sealing process”and “The modification of LDH sealing film via PF” …. detail the processes further. (add pictures if possible)

Response: The reaction principles of the LDH sealing process and PF modification are further expounded in section of “3.2 The characterization of CSA-LDH film” and Figure 4, respectively.

  1. Evaluation of SEM images should be increased. (Figure 3 and 4).

Response: As you suggested, the relevant evaluations have been added in sections of “3.2 The characterization of CSA-LDH film” and “3.3 The characterization of CSA-LDH-PF film”.

  1. …… the CSA-LDH-PF sample exhibits the maximal resistance, and the barrier layer appears in the low frequency segment……More explain, Why?

Response: We apologize for such unclear expressions. Detailed explanations of Figure 8c-d are added in section of “3.4 Electrochemical measurements”.

  1. …….Hydrophilic CSA-LDH film with surface hydroxyl groups generated during the sealing process further decreased the contact angle to 4.6° (Figure 9c). However, after modification by PF, the surface groups are changed to hydrophobic CF3(CF2)5(CH2)2- with low surface energy [30] and……. This section should be more detailed. (Specific explanations required)

Response: Thank you for your advice. More specific explanations have been in section of “3.5 Contact angle tests” to expound the changes of the sample surface.

  1. Conclusions section should be enriched.

Response: The conclusions section has been enriched as you suggested.

  1. More literature studies should be added to the introduction and other sections (DOIs given below).

DOI-1   https://doi.org/10.1007/s13369-021-06243-w  (about different studies)

DOI-2  https://doi.org/10.35193/bseufbd.1075980  (about different studies)

DOI-3  https://doi.org/10.53525/jster.1174394  (about welding aluminum alloys)

Response: As you suggested, the literature mentioned above are cited as References 9, 10 and 3, respectively.

Round 2

Reviewer 1 Report

It seems the authors have made the necessary changes requested

Reviewer 2 Report

Manuscript ID materials-1982123 entitled " Fluorinated siloxane modified layered double hydroxide sealing film to enhance the corrosion resistance of anodic oxide film of fricition stir welding joint of aluminum alloys" for journal of Materials has been reviewed.

Title of manuscript- Suitable with contex

-Abstract clearly presents objects methods and results.

-Scientific methods are adequately used.

-Terminology is adequate.

-Results are clearly presented .

-Conclusions are logically derived from the data presented.

-Key words are adequate.

-References are appropriate.

The authors have made the necessary corrections.

The article is suitable for publication.

Decision- Accept

 --------------------------------------

Congratulations to the authors.

I wish the authors success in their future academic studies.

Kind regards.